# Structure and dynamics underlying elementary ligand binding events in human pacemaking channels

Marcel P Goldschen-Ohm[1†], Vadim A Klenchin[1†], David S White[1,2], John B Cowgill[1], Qiang Cui[3], Randall H Goldsmith[2], Baron Chanda[1,4*]

[1]Department of Neuroscience, University of Wisconsin-Madison, Madison, United States; [2]Department of Chemistry, University of Wisconsin-Madison, Madison, United States; [3]Department of Chemistry and Theoretical Chemistry Institute, University of Wisconsin-Madison, Madison, United States; [4]Department of Biomolecular Chemistry, University of Wisconsin-Madison, Madison, United States

**Abstract** Although molecular recognition is crucial for cellular signaling, mechanistic studies have relied primarily on ensemble measures that average over and thereby obscure underlying steps. Single-molecule observations that resolve these steps are lacking due to diffraction-limited resolution of single fluorophores at relevant concentrations. Here, we combined zero-mode waveguides with fluorescence resonance energy transfer (FRET) to directly observe binding at individual cyclic nucleotide-binding domains (CNBDs) from human pacemaker ion channels critical for heart and brain function. Our observations resolve the dynamics of multiple distinct steps underlying cyclic nucleotide regulation: a slow initial binding step that must select a 'receptive' conformation followed by a ligand-induced isomerization of the CNBD. X-ray structure of the apo CNBD and atomistic simulations reveal that the isomerization involves both local and global transitions. Our approach reveals fundamental mechanisms underpinning ligand regulation of pacemaker channels, and is generally applicable to weak-binding interactions governing a broad spectrum of signaling processes.

*For correspondence: chanda@wisc.edu

[†]These authors contributed equally to this work

## Introduction

Most cellular signaling pathways require or are modulated by the binding of small molecules to integral proteins. However, our understanding of the dynamic events involved in these molecular recognition processes comes primarily from inferences based on downstream activity initiated by binding, or ensemble measures that average over and thereby obscure the underlying mechanistic steps. In contrast, single-molecule observations reveal dynamics and heterogeneity of conformational transitions that are otherwise averaged over in ensemble measurements, and thus are a means to probe specific molecular transitions providing important clues to the physical basis for binding (*Joo et al., 2008*; *Csermely et al., 2010*; *Greives and Zhou, 2014*; *Guo and Zhou, 2016*; *Ruiz and Karpen, 1997*; *Miller, 1997*). Single-molecule approaches have provided mechanistic insight in many areas: for example, patch-clamp recordings from single ion channels reveal the network of states that underlie gating of the central pore (*Lape et al., 2008*; *Mukhtasimova et al., 2009*; *Colquhoun and Lape, 2012*; *Purohit et al., 2014*), whereas optical techniques such as single-molecule FRET (smFRET) allow tracking of conformations and structural movements of individual domains (*Akyuz et al., 2015*; *Cooper et al., 2015*; *Wang et al., 2014*; *Vafabakhsh et al., 2015*; *Landes et al., 2011*; *Wang et al., 2016*). However, similar resolution of the fundamental mechanisms underlying individual ligand binding events that initiate or modulate downstream domain

**eLife digest** Certain cells in the heart and brain show rhythmic bursts of electrical activity. Such electrical activity is a caused by ions moving in or out of the cells though a number of ion channel proteins in the cell surface membrane. The voltage across this cell membrane regulates the activity of these so-called pacemaking channels, and so do small molecules like cAMP. Nevertheless, it remained poorly understood how cAMP binding altered how the channels work. This was because researchers had been unable to unambiguously resolve the early binding events, because the available techniques were too limited.

Goldschen-Ohm, Klenchin et al. have now overcome these technical limitations and observed individual molecules of cAMP (which had been first labeled with a fluorescent tag) binding to the relevant parts of a pacemaking channel from humans. This approach revealed that the binding process happens via a sequence of discrete steps. First, cAMP selectively binds when the binding site of the ion channel adopts a specific shape, called its "receptive" state. Second, part of the protein rotates which changes the shape of the binding site and traps the bound cAMP in place. The trapped molecule is not released until the binding site reverts to its previous shape.

These new findings provide the groundwork for future studies to dissect how different parts of pacemaking channels change shape and interact to control these channels' activities.

movements and pore gating are lacking, primarily due to technical challenges in imaging with both sufficient temporal resolution and at concentrations necessary to drive many physiologically relevant recognition processes.

For fluorescence based approaches (*Funatsu et al., 1995*), a major challenge hampering resolution of single binding events with low affinity is the diffraction limit of light microscopy. At the high concentrations necessary to drive these binding reactions, the number of fluorescent ligands within the diffraction-limited excitation volume becomes appreciable, thereby obscuring resolution of individual fluorophores. Unfortunately, many physiologically relevant recognition processes have affinities in the micromolar range, which precludes single-molecule resolution with traditional microscopy techniques including total internal reflection (TIRF) or confocal microscopy.

To observe micromolar affinity binding events at single molecules, we used zero-mode waveguide (ZMW) nanofabricated devices (*Levene et al., 2003*; *Zhu and Craighead, 2012*). ZMWs limit optical excitation to a sub-diffraction-limited volume such that even at micromolar concentrations there are sufficiently few ligands excited that binding of a single fluorophore in the excitation volume can be resolved. As a notable exception to the overall lack of single-molecule binding observations for physiological processes, ZMWs have been used to great effect to study translation events at individual ribosomes and single-molecule electrochemistry, and have enabled single-molecule genomic sequencing (*Uemura et al., 2010*; *Korlach et al., 2010*; *Zhao et al., 2013*).

Here, we combined ZMWs with smFRET to resolve individual specific binding events of a fluorescent cyclic nucleotide derivative (fcAMP) (*Kusch et al., 2010*) with micromolar affinity for its receptor CNBD from hyperpolarization-activated cyclic nucleotide-gated (HCN) channels critical for oscillatory neuronal activity in the brain and pacemaking in the heart. Although binding of cyclic nucleotides is known to enhance HCN voltage-dependent activation, the mechanisms that underlie this regulation remain unclear. Previous studies of cyclic nucleotide (e.g. cAMP) regulation have relied primarily on ensemble channel currents (*Chen et al., 2007*), or more recently on ensemble fluorescence from fcAMP (*Kusch et al., 2012*; *Benndorf et al., 2012*; *Thon et al., 2015*), to deduce the dynamics of cyclic nucleotide association. Although fcAMP provides a more direct measure of binding than does downstream pore current, both measurements reflect ensemble-averaged data that obscures resolution of the individual steps involved in the binding process. Resolving the dynamics of these steps is important because it provides a rationale for assigning the effect of specific perturbations to distinct mechanistic steps – an invaluable tool for deconstructing the pathway by which binding is transduced to functional changes elsewhere such as at the pore gate.

To resolve ambiguity in current ensemble-based models of cyclic nucleotide association, we dissected the intrinsic binding dynamics at single molecules to reveal that cAMP binding involves

multiple conformational transitions: an initial binding step that is appreciably slower than expected for a diffusion-limited encounter complex, partly due to selection of the 'receptive' conformation, and a subsequent ligand-induced isomerization of the CNBD. Our single-molecule observations in both monomeric and tetrameric CNBD complexes, in conjunction with the first unique X-ray structure of the unliganded CNBD and molecular dynamics (MD) simulations, resolve the dynamic mechanisms underlying cyclic nucleotide association at HCN channels to a level of unprecedented detail.

## Results

### Single-molecule fluorescent cAMP binding

For optical tracking of the ligand we used a fluorescent cyclic nucleotide conjugate (fcAMP) that modulates HCN2 channel function in a very similar manner to native non-conjugated cAMP (*Kusch et al., 2010*). The purified CNBD from HCN2 channels was engineered to contain a single accessible cysteine residue (C508A/C584S/E571C) near the binding pocket that was specifically labeled with a fluorescent FRET acceptor (*Figure 1A–B*), hereafter referred to simply as CNBD unless stated otherwise. The only other cysteine residue C601 is buried in the hydrophobic core of the CNBD as indicated by both crystal structures (*Zagotta et al., 2003*; *Lolicato et al., 2011*) and our inability to label this position with a maleimide reactive fluorophore. Ensemble fluorescence anisotropy revealed that both native and mutated CNBDs exhibited similar affinity for fcAMP (*Figure 1—figure supplement 1A*). Efficient FRET due to specific binding of fcAMP to the acceptor-labeled CNBD was confirmed in bulk solution (*Figure 1—figure supplement 1B*).

For single-molecule measurements, isolated acceptor-labeled CNBDs were sparsely deposited in ZMW arrays bathed in concentrations of fcAMP ranging from 0.1–10 μM. Donor and acceptor emission was simultaneously recorded from arrays of distinct ZMWs in response to alternating direct excitation of either donor or acceptor (*Santoso et al., 2008*) (*Figures 1C–D* and *2A*, *Figure 1—figure supplement 2*). Representative fluorescence time traces for fcAMP binding to single CNBD molecules are shown in *Figure 2*. Unlike typical FRET experiments where both the donor and acceptor are immobilized, the donor (fcAMP) was free to diffuse in and out of the excitation volume surrounding the immobile acceptor on the CNBD. Because the time for free fcAMP to diffuse across the excitation volume (~1 ms) is extremely short compared to the duration of a single image frame (100 ms), we did not resolve the diffusion of fcAMP to or from the CNBD, but only observed increased fluorescence when fcAMP remained within the excitation volume for a time period comparable to or longer than the frame duration, as when bound to the CNBD. In addition, observation that excitation of a bound donor resulted primarily in an increase in acceptor emission, with little to no observable increase in donor emission, suggests that FRET efficiency between bound fcAMP and the nearby acceptor was close to 100%. This response is not due to blinking of a weakly excited acceptor during donor excitation because direct excitation of the acceptor on alternating frames resulted in stable acceptor emission intensities until the acceptor bleached. Furthermore, whenever fcAMP remained bound at the time of acceptor bleaching, we immediately observed emission from the donor. The FRET signal shown in *Figure 2B* could also be abolished by competition with excess non-fluorescent cAMP (*Figure 2—figure supplement 1*), and thus represents a time-dependent binary readout for specific fcAMP binding at single molecules.

*Figure 2B* provides a visual illustration of both the challenge of resolving single binding events at micromolar concentrations and the advantages of ZMWs. First, at the concentrations required to drive fcAMP binding, fluorescence from individual bound donors cannot be reasonably resolved apart from the freely diffusing donors within a diffraction-limited spot using traditional approaches such as confocal or TIRF microscopy. In contrast, ZMWs reduce the excitation volume beyond that achievable with TIRF, such that even at micromolar concentrations a sufficiently small number of fluorophores are excited that single bound donors can be detected (*Levene et al., 2003*; *Zhu and Craighead, 2012*) (e.g. donor fluorescence after acceptor bleaching in *Figure 2B*). The combination of ZMWs and smFRET can further reduce background fluorescence from free donor and yet allow resolution of single binding events while limiting artifacts due to nonspecific binding at the surface (e.g. smFRET in *Figure 2B*). Furthermore, alternating between donor and acceptor excitation provides explicit information on the number of molecules within each ZMW in the form of acceptor bleach steps (e.g. acceptor fluorescence in *Figure 2B*).

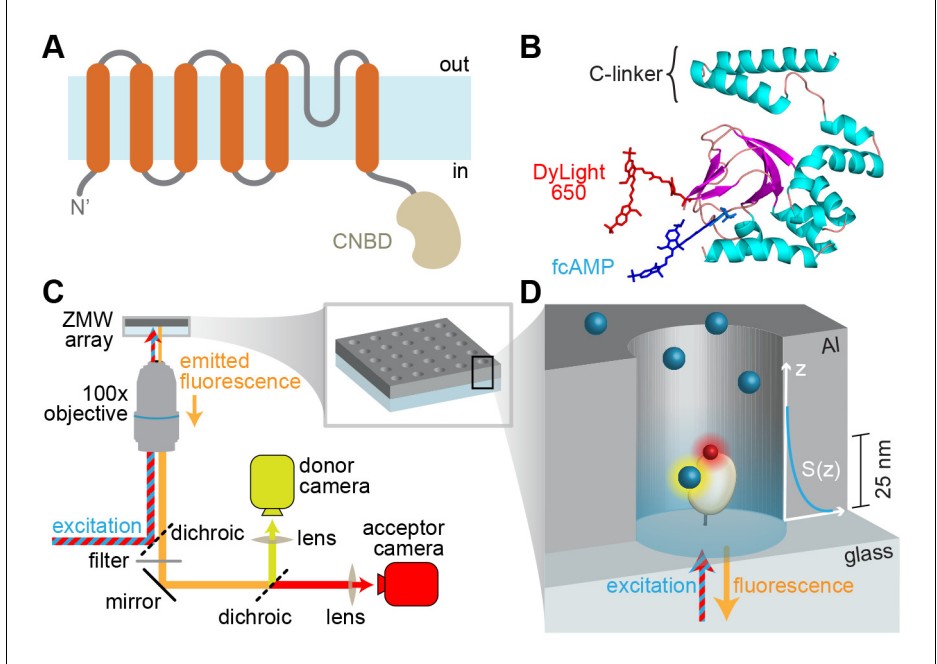

**Figure 1.** Imaging ligand binding to single molecules in ZMWs. (**A**) HCN2 channel subunit transmembrane topology is homologous to canonical voltage-gated potassium channel subunits with the exception of a CNBD following the pore lining S6 helix. (**B**) Isolated CNBD colored by secondary structure with fluorescent acceptor DyLight 650 maleimide attached at position E571C and bound donor fcAMP (cAMP + DyLight 547) shown. (**C**) ZMW smFRET imaging setup (inset shows a rendering of an array of ZMWs) and (**D**) cartoon of an individual ZMW with a single fcAMP-bound acceptor-labeled CNBD tethered to the optical surface within the aluminum (Al) nanopore (drawing is not to scale). Bound fcAMP (blue sphere) near the bottom of the ZMW is directly excited and emission from the acceptor (red sphere) on the CNBD due to FRET is observed. In contrast, freely diffusing fcAMP molecules are shown near the top of the ZMW where they are outside of the effective near-field observation volume, and thus not observed. The scale bar to the right of the ZMW indicates a typical length constant for the exponentially decaying observation volume S(z), which was estimated by extrapolating reported values for various ZMW diameters (*Levene et al., 2003*).

The following figure supplements are available for figure 1:

**Figure supplement 1.** Bulk solution fluorescence imaging of fcAMP binding.

**Figure supplement 2.** Imaging ZMW arrays.

## A conformational change following binding

The probability of being in a bound state across all time points and all molecules as a function of fcAMP concentration was described by a binding curve with an apparent dissociation constant of 1.5 μM (*Figure 3A*). The similar affinities obtained with both single-molecule and bulk solution anisotropy measurements (2.3 μM; *Figure 1—figure supplement 1A*) suggest that non-specific interactions with the ZMW surface are weak, and that our observations reflect inherent dynamics of monomeric isolated CNBDs.

As expected for a ligand binding reaction, distributions of unbound dwell times derived from fitting idealized smFRET traces shifted to shorter durations with increasing fcAMP concentration, whereas the bound time distributions were essentially independent of fcAMP concentration (*Figure 3B*). Maximum likelihood estimates of unbound and bound dwell time distributions required two exponential components each (*Figure 3C*, *Figure 3—figure supplement 1*), whereas monoexponential distributions resulted in poor descriptions of the data, and triexponential distributions reduced to biexponential distributions in all cases (i.e. one component had zero amplitude). The probability that any of the observed unbound or bound time distributions were monoexponential as

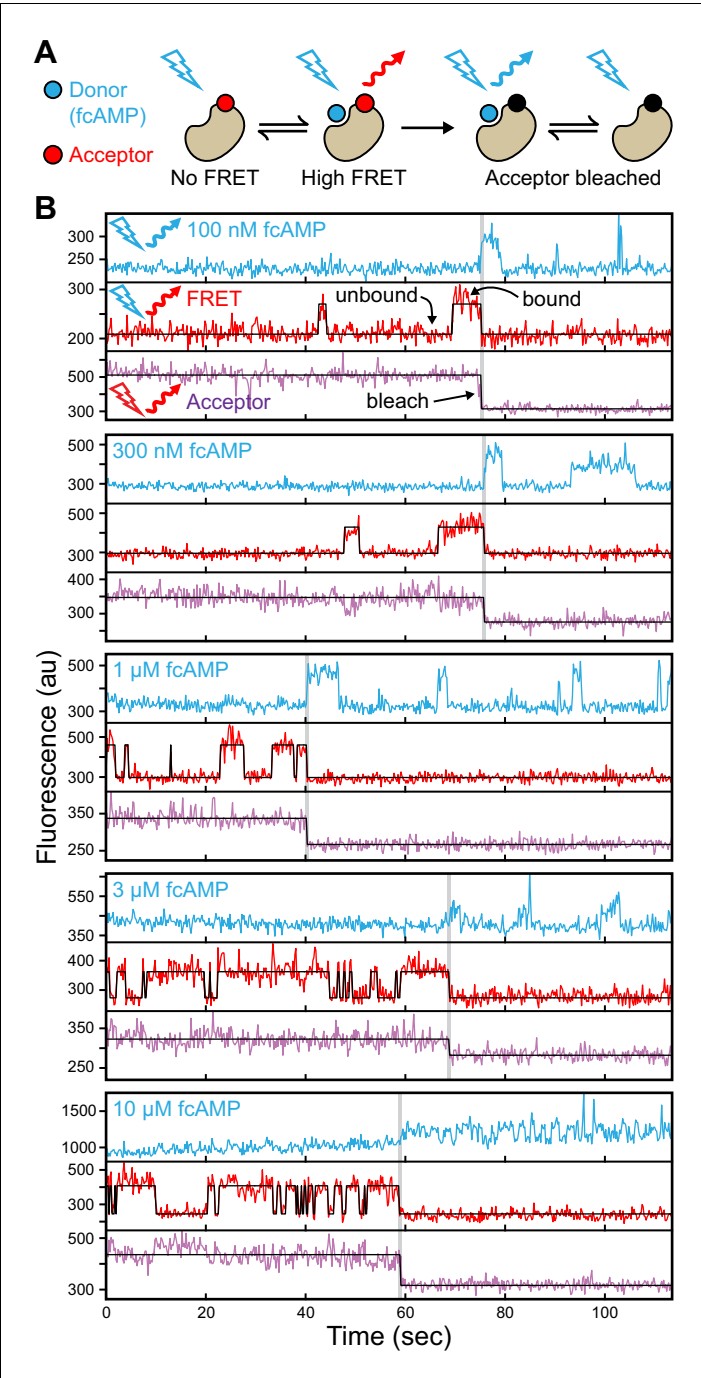

**Figure 2.** Singe-molecule FRET imaging of binding dynamics at micromolar concentrations. (**A**) Cartoon depicting smFRET during fcAMP binding. Direct excitation of the donor fcAMP results in stimulated emission from the acceptor on the CNBD due to efficient FRET while the donor is bound up until the acceptor bleaches, after which only emission from the donor is observed. (**B**) Single-molecule fluorescence time series for fcAMP binding to individual acceptor-labeled CNBDs within ZMWs. Simultaneous emission from donor (blue) and acceptor (red) upon donor excitation at 532 nm was interleaved every other frame with emission from acceptor (magenta) upon direct excitation at 640 nm. Acceptor fluorescence for both excitation conditions is overlaid with the idealized time series (black). fcAMP concentration is indicated in the upper left of each donor fluorescence trace.

The following figure supplement is available for figure 2:

**Figure supplement 1.** Specific fcAMP binding at single molecules.

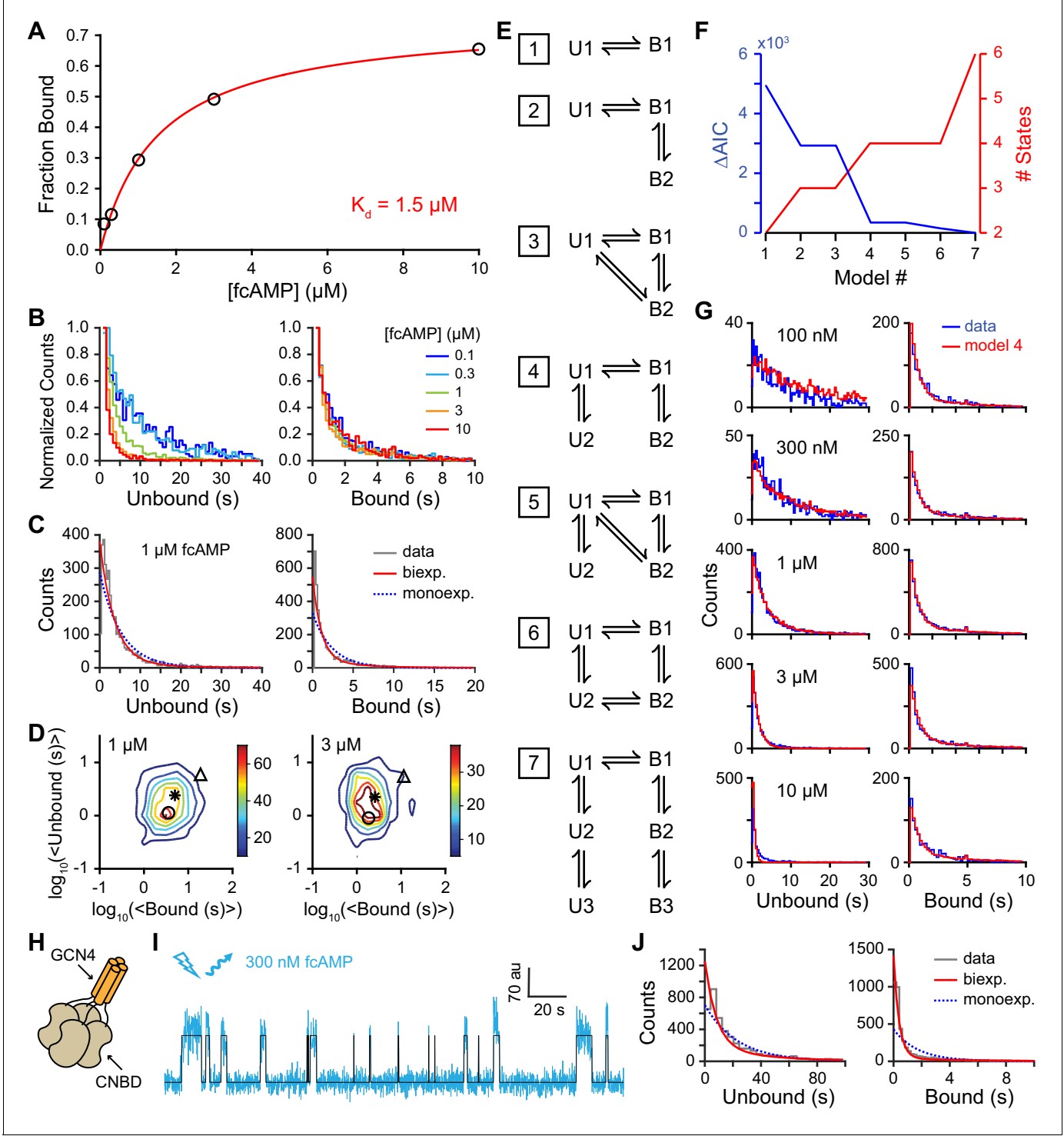

**Figure 3.** Single-molecule cyclic nucleotide binding dynamics at HCN2 CNBDs. (A) Bound probability from the total fraction of time spent bound for all single molecules as a function of fcAMP concentration fit with the equation $B_{max}/(1+K_d/[fcAMP])$, where $B_{max} = 0.75$ is the maximal bound probability and $K_d = 1.5\ \mu M$ is the apparent dissociation constant. (B, C) Histograms of unbound and bound single-molecule dwell time distributions for events from all molecules combined for (B) various concentrations of fcAMP and (C) distributions for 1 µM fcAMP overlaid with maximum likelihood estimates for monoexponential (blue dashed) and biexponential (red) distributions. Exponential fits with estimated parameters and confidence intervals for all tested fcAMP concentrations are shown in *Figure 3—figure supplement 1*. (D) Contour plots of two dimensional histograms for the average bound time versus average unbound time per molecule at several fcAMP concentrations. Color bar denotes number of molecules. Symbols denote time constants from maximum likelihood biexponential fits (open circle and triangle) and their amplitude-weighted average (asterisk). Similar contours for all

*Figure 3 continued on next page*

*Figure 3 continued*

tested fcAMP concentrations are shown in *Figure 3—figure supplement 2*. (**E**) Kinetic models between unbound (U*) and bound (B*) states. The model ID number is indicated to the left of each model. Optimized rate constants are given in *Table 1*. (**F**) Differences in the Akaike Information Criteria (AIC) for optimized models shown in (**E**). (**G**) Comparison of observed dwell time histograms with simulated data from model 4. Histogram abscissas for (**B**), (**C**) and (**G**) were truncated to facilitate visualization of the faster components. (**H**) Cartoon illustrating a tetrameric CNBD complex formed by appending tetramerizing GCN4 coiled-coil to the N-terminus. (**I**) Example fluorescence time series (blue) for fcAMP binding events at a single tetramer in a ZMW overlaid with the idealized trace (black). At 300 nM fcAMP, the probability that more than one CNBD is bound at any given time is low. (**J**) Dwell time distributions for the first binding step in CNBD tetramers. Bound lifetimes are biexponentially distributed as for monomeric CNBDs.

The following figure supplements are available for figure 3:

**Figure supplement 1.** Maximum likelihood estimates of biexponential parameters for single-molecule dwell time distributions.

**Figure supplement 2.** Dwell time correlations within single molecules.

**Figure supplement 3.** CNBD tetramer.

**Figure supplement 4.** Binding with and without smFRET.

opposed to biexponential was less than 0.001 as determined from a $\chi^2$ distribution with two degrees of freedom corresponding to twice the difference in their log likelihoods. The biexponentially distributed dwell times suggest that either each CNBD can sample two unbound and two bound metastable conformations with two corresponding rate constants for each process, or our data is comprised of a bimodal population of CNBD molecules, each with distinct single-step binding dynamics. In the latter case, the correlation between the average bound versus unbound time for each isolated CNBD should cluster into two groups corresponding to the time constants for binding and unbinding in each population. However, at all observed fcAMP concentrations the per molecule average unbound versus bound time distribution was centered around the weighted average of the two pairs of time constants and broadly distributed between them (*Figure 3D*, *Figure 3—figure supplement 2A*). This behavior suggests dynamic heterogeneity, where each individual CNBD molecule was able to repeatedly interconvert between two unbound (U1 and U2) and two bound (B1 and B2) configurations during a typical recording (~1 min.). Furthermore, individual bound durations did not depend on the preceding unbound duration, consistent with sequential models of binding (*Figure 3—figure supplement 2B–D*).

To explore the functional dynamics underlying our data, we compared likelihoods of several kinetic models using hidden Markov modeling (HMM) (*Bronson et al., 2009*; *Nicolai and Sachs, 2013*) (*Figure 3E*; *Table 1*). Models were globally optimized for smFRET time series from all

**Table 1.** Kinetic model rate constants. Optimized rate constants ($s^{-1}$ or $M^{-1}s^{-1}$) for models shown in *Figure 3E*. U* and B* denote unbound and bound states, respectively.

| Model | U1→ B1 | B1→ U1 | B1→B2 | B2→B1 | U1→U2 | U2→U1 | U1→B2 | B2→U1 |
|---|---|---|---|---|---|---|---|---|
| 1 | $1.3 \times 10^5$ | 0.34 | - | - | - | - | - | - |
| 2 | $1.4 \times 10^5$ | 0.91 | 0.52 | 0.31 | - | - | - | - |
| 3 | $1.3 \times 10^5$ | 0.98 | 0.49 | 0.23 | - | - | $0.10 \times 10^5$ | 0.04 |
| 4 | $2.3 \times 10^5$ | 0.95 | 0.51 | 0.31 | 0.04 | 0.15 | - | - |
| 5 | $2.2 \times 10^5$ | 1.00 | 0.49 | 0.25 | 0.04 | 0.15 | $0.14 \times 10^5$ | 0.03 |
| 6 | $2.4 \times 10^5$ | 1.00 | 0.48 | 0.27 | 0.01 | 0.04 | - | - |
| 7 | $2.8 \times 10^5$ | 1.11 | 0.85 | 0.56 | 0.17 | 0.55 | - | - |
| Model | U2→B2 | B2→U2 | U2→U3 | U3→U2 | B2→B3 | B3→B2 | | |
| 6 | $0.24 \times 10^5$ | 0.02 | - | - | - | - | | |
| 7 | - | - | 0.02 | 0.07 | 0.03 | 0.08 | | |

molecules and fcAMP concentrations and ranked according to their Akaike information criterion (AIC) (*Akaike, 1974*) (*Figure 3F*). Consistent with dwell time distributions discussed above, models with two unbound (U1, U2) and two bound (B1, B2) states performed better than models with fewer states, whereas adding additional unbound and bound states did not improve the likelihood. Remarkably, all of the models with two bound states converged to a similar set of rate constants governing an initial binding step followed by a subsequent first order transition (*Table 1*). The most direct interpretation of this finding is that the CNBD undergoes a reversible conformational change between distinct cAMP-bound conformations. Importantly, simulated data from such a model reproduced the experimentally observed dwell time distributions (*Figure 3G*).

To test whether the binding dynamics that we observed in monomeric CNBDs were reflective of their functional dynamics in tetrameric channels, we generated CNBD tetramers using a GCN4 tetramerization motif N-terminal of the CNBD to mimic the S6 helices of the channel pore (*Figure 3H*, *Figure 3—figure supplement 3*). Although neither monomeric nor tetrameric constructs are attached to a true channel pore, the tetramer is a useful construct for testing the role of inter-CNBD interactions that are likely to play a role in channel regulation (e.g. cAMP induces tetramerization of isolated CNBDs even in the absence of a channel pore [*Lolicato et al., 2011*]). To resolve the dynamics of the first of four binding steps, we observed fluorescence from bound fcAMP at tetramers deposited in ZMWs bathed in 300 nM fcAMP (*Figure 3I*). At this concentration, single binding events can be resolved directly from the emission intensity time course of fcAMP (see *Figure 2B*, *Figure 3—figure supplement 4*), and thus acceptor labels on tetramers were bleached prior to recording fluorescence from direct excitation of fcAMP (in contrast, smFRET was needed to resolve binding at higher fcAMP concentrations in monomers; see *Figure 2B*). Analysis of the first binding step in CNBD tetramers confirms that the biexponential nature of the bound time distributions are an inherent property of CNBDs both in isolation and in tetrameric complexes (*Figure 3J*). Taken together, dwell time analysis for monomers and tetramers and HMM modeling show that following binding, individual CNBDs interconvert between two bound conformations with distinct lifetimes (as discussed below).

## The nature of the ligand-induced conformational change

To gain structural perspective into the observed single-molecule dynamics, we turned to X-ray crystallography. Several structures of the cAMP-bound (holo) CNBD from eukaryotic HCN channels have been solved (*Zagotta et al., 2003*; *Xu et al., 2010*; *Lolicato et al., 2011*). However, a unique conformation of the unliganded (apo) CNBD from HCN channels has been recalcitrant to crystallization, with the only reported crystal of the apo form containing a conformation essentially identical to the ligand-bound form (*Taraska et al., 2009*) (less than 1 Å RMSD difference between the two). Distance constraints from nuclear magnetic resonance (NMR) spectroscopy has been used to generate a structural model of the apo form of the HCN2 CNBD (*Saponaro et al., 2014*), but this structure has many features that are not consistent with other cyclic nucleotide binding domains found in the Protein Data Bank (*Clayton et al., 2004*; *Kim et al., 2005*; *Schunke et al., 2011*). These discrepancies include complete unfolding of the conserved P-helix in the phosphate binding cassette (PBC) and an unexpectedly high degree of conformational flexibility in the β-roll domain. Hence, we crystallized and solved the structure of the apo HCN2 CNBD as part of a fusion protein with the maltose binding protein (MBP) in order to provide a high-resolution counterpart to the existing holo crystal structures. An overview of the structure and representative electron densities are presented in *Figure 4—figure supplement 1*, and crystallographic statistics are given in *Table 2*. The crystallized construct lacks the first three α-helices of the C-linker, making it unable to tetramerize. As discussed below, this was a key factor because tetramers favor a closed conformation typical of the nucleotide-bound CNBD. The majority of the crystal contacts are formed by MBP, which is significantly more ordered as compared to the CNBD (mean B-factors for MBP = 22.9 Å$^2$, and CNBD = 43.2 Å$^2$), making it unlikely that crystal packing interactions contribute significantly to the observed structure of the HCN2 CNBD (*Figure 4—figure supplement 3*). As discussed below, the overall conformation of the CNBD monomer is substantially the same as that found by NMR (*Saponaro et al., 2014*), further suggesting that the structure observed in the crystal is not appreciably altered by crystal contacts or fusion with MBP.

The large-scale structural changes between apo and holo structures involve both N- and C-terminal α-helical fragments that undergo a rigid body rotation around hinges that correspond to the

**Table 2.** Crystallographic statistics.

| Data collection | |
| --- | --- |
| Space group | P2$_1$ |
| Unit cell dimensions | |
| a, b, c (Å) | 61.5, 42.0, 198.4 |
| α, β, γ (°) | 90, 90.9, 90 |
| Resolution (Å) | 24.82–2.07 (2.11–2.07) |
| Unique reflections | 62519 |
| Redundancy[*] | 3.4 (3.4) |
| Average [I/s] [*] | 7.6 (2.2) |
| Completeness (%)[*] | 99.9 (100) |
| R$_{merge}$ (%)[*] | 11.5 (66.7) |
| Refinement | |
| Number of atoms | |
| protein | 7913 |
| maltose | 46 |
| solvent | 381 |
| R$_{work}$ (%)[*] | 18.0 (21.6) |
| R$_{free}$ (%)[*] | 22.1 (26.0) |
| Twin fraction | 0.122 |
| Average B-factors, (Å$^2$) | |
| protein (MBP), protein (HCN2) | 22.9, 43.2 |
| solvent | 28.9 |
| R.M.S. deviations, bond angles (°) | 1.322 |
| R.M.S. deviations, bond lengths, (Å) | 0.009 |
| Ramachandran plot (%) | |
| favored | 98 |
| allowed | 2 |

[*] Values in parentheses are for the outer resolution shell.

points where they emerge out of the rigid β-roll cage (*Figure 4A–B*). The result of these rotations is the closed conformation in which the C-helix swings toward the ligand binding site and caps bound cAMP. The core β-roll structure is essentially invariant to the presence of ligand (*Figure 4C*). Strikingly, when the N- and C-terminal helical termini are treated as a single module, they too can be well superposed between the apo and holo conformations (*Figure 4D*). This indicates that the hinge-like motions of both termini (*Figure 4E–F*) are coordinated in a way that the α-helical structures combine to form a single rigid body domain. The coordinated movement of N- and C-terminal domains is necessitated by the fact that motion of the C-helix toward cAMP introduces numerous steric clashes with the apo conformation of the N-terminal domain (*Figure 4G*). These clashes are relieved in the holo conformation by an upward movement of the N-terminus by as much as 7 Å (compare *Figure 4A and B*), ultimately placing the N-terminal C-linker into a tetramerization-competent state. This conformational coupling between the two α-helical domains enables cAMP-induced structural changes to be transmitted across the entire CNBD toward the gating pore. This idea is illustrated schematically for a pair of CNBDs in *Figure 4H*.

The trigger that induces these large-scale conformational changes upon cAMP binding is the localized conformational change in the PBC (*Berman et al., 2005*) whose P-helix moves toward the ligand and undergoes a subtle transition from a mostly 3$_{10}$-helix to a mostly α-helix (*Figure 4C*, *Table 3*). In contrast to the reported NMR apo structure where the P-helix residues are fully

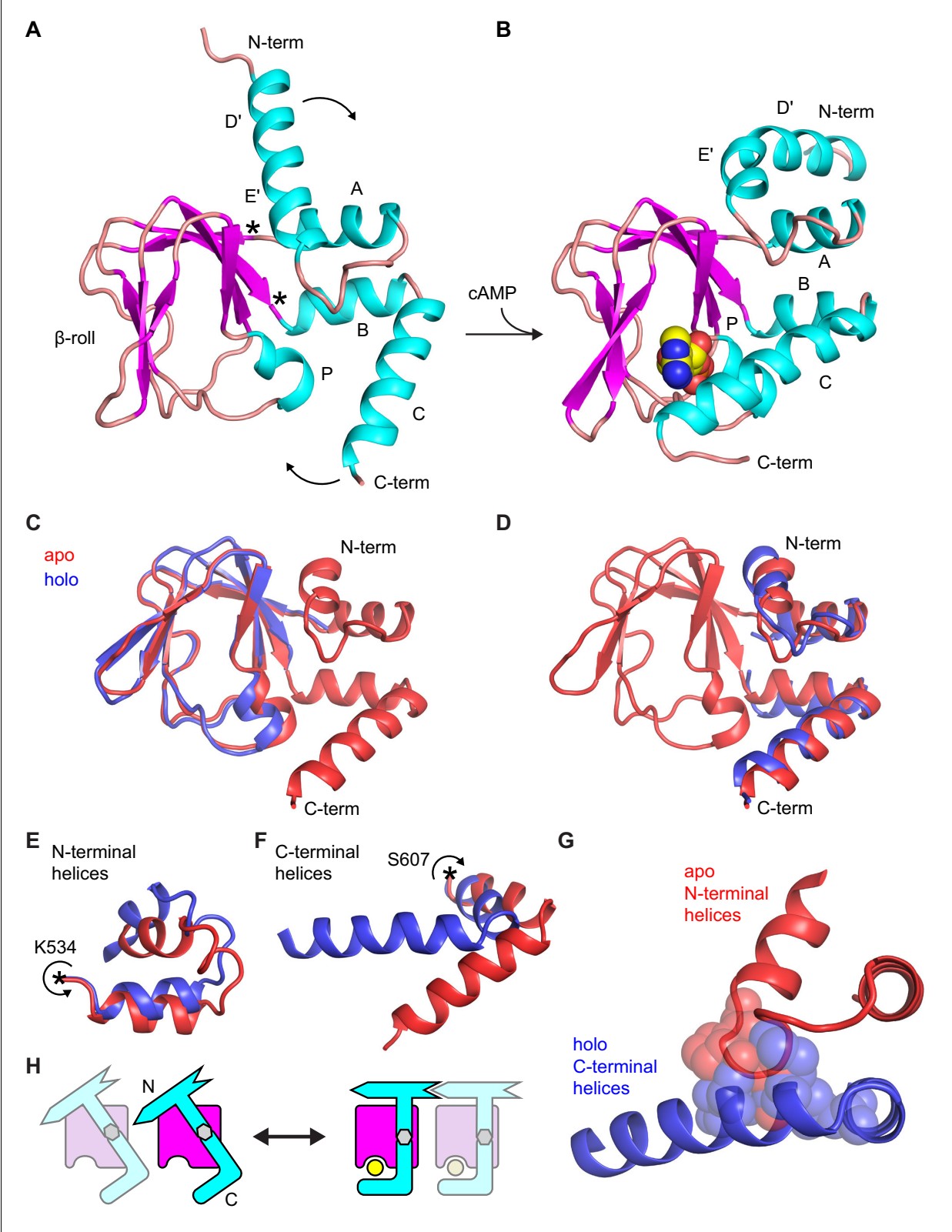

**Figure 4.** Comparison of apo and holo forms of the HCN2 CNBD. (A) X-ray crystal structures of apo (this work) and (B) holo (PDB 3U10) conformations of the HCN2 CNBD colored according to protein secondary structure. Bound cAMP in the holo structure is shown as spheres. Helical domains are labeled D', E', A, B, C and P as was previously done for the holo structure (*Zagotta et al., 2003*). (C–D) Apo structure (red) overlaid with either the β-roll domain (C) or the α-helical termini (treated as a single domain) (D) of the holo structure (blue). The apo and holo structures superimpose with an

*Figure 4 continued on next page*

*Figure 4 continued*

RMSD of 0.49 Å over their β-roll domains, or 2.04 Å over their α-helical terminal domains (N-terminal helix excludes the initial D' segment). Hinge-like rotations that account for bulk of conformational changes between apo and holo structures: (E) N-terminal helical fragment (residues 508–534), (F) C-terminal helical fragment (residues 607–634). (G) Steric clashes between holo conformation of the C-terminus (blue) and apo conformation of the N-terminus (red). Residues that would clash are illustrated as spheres (M515, P516, L517 and L615, M621, A624, F625). (H) Cartoon illustrating the cAMP-induced rotation of the α-helical domains (cyan) about the rigid β-roll cage (magenta) that both caps the bound ligand (yellow) and places the N-terminal region in a favorable state for coordinating intersubunit interactions with neighboring CNBDs (indicated by a faded CNBD).

The following figure supplements are available for figure 4:

**Figure supplement 1.** Overview of the X-ray structure.

**Figure supplement 2.** Comparison of the X-ray and NMR structures of the apo CNBD.

**Figure supplement 3.** Molecular packing in the crystal of apo MBP-HCN2.

unfolded, the P-helix remains helical in the X-ray apo structure reported here (*Figure 4—figure supplement 2*), consistent with apo versus holo crystal structures of CNBDs from both MlotiK1 (*Clayton et al., 2004*; *Schunke et al., 2011*) and the regulatory subunit of PKA (*Kim et al., 2005*).

Although crystal X-ray and solution NMR structures are in agreement on large-scale movements of the α-helical termini, they substantially differ in large stretches of residues that comprise the β-roll (*Figure 4—figure supplement 2*). In the apo-structure presented here, the β-roll is essentially superimposable to those observed in the different holo structures solved under a variety of conditions (*Zagotta et al., 2003*; *Taraska et al., 2009*; *Saponaro et al., 2014*). Thus, the β-roll domain, save for the P-helix and the flexible T566-E571 loop, represents a semi-rigid cage built around the cAMP binding pocket.

## Molecular dynamics simulations

Considering the slow time scale associated with the U1-B1 and B1-B2 transitions, and the fact that there is little explicit experimental characterization for the B1 state, we do not anticipate that molecular dynamics (MD) simulations for these transitions are warranted. However, MD simulations are useful for further understanding how the ligand (cAMP) stabilizes the holo structure at both local and global scales. Along this line, we have carried out microsecond MD simulations for the holo structure with cAMP removed from the binding site, and for the apo structure with cAMP manually docked into the binding site; cAMP-bound holo simulation and apo simulation without any ligand were also carried out to serve as controls and references (*Figure 5*). These simulations are not designed to probe the conformational transition pathway; indeed, the holo simulation with cAMP removed does not correspond to a process in the proposed kinetic model (*Figure 6*). Instead, by monitoring structural responses to a change in the ligation state, we are able to identify structural motifs that are most directly stabilized by cAMP binding; this is not straightforward to accomplish by examining the static crystal structures alone, especially when allosteric effects are implicated.

**Table 3.** cAMP-dependent change in H-bonding within PBC.

| Apo (this work) | | | | Holo (PDB 3U10) | | | |
|---|---|---|---|---|---|---|---|
| carbonyl | amide | *d*, Å | pattern | carbonyl | amide | *d*, Å | pattern |
| G581 | C584 | 3.42 | i + 3 | G581 | — | — | — |
| E582 | L585 | 3.02 | i + 3 | E582 | L586 | 3.07 | i + 4 |
| I583 | L586 | 3.10 | i + 3 | I583 | L586 | 3.10 | i + 3 |
| I583 | T587 | 2.71 | i + 4 | I583 | T587 | 2.71 | i + 4 |
| C584 | — | — | — | C584 | R588 | 2.91 | i + 4 |

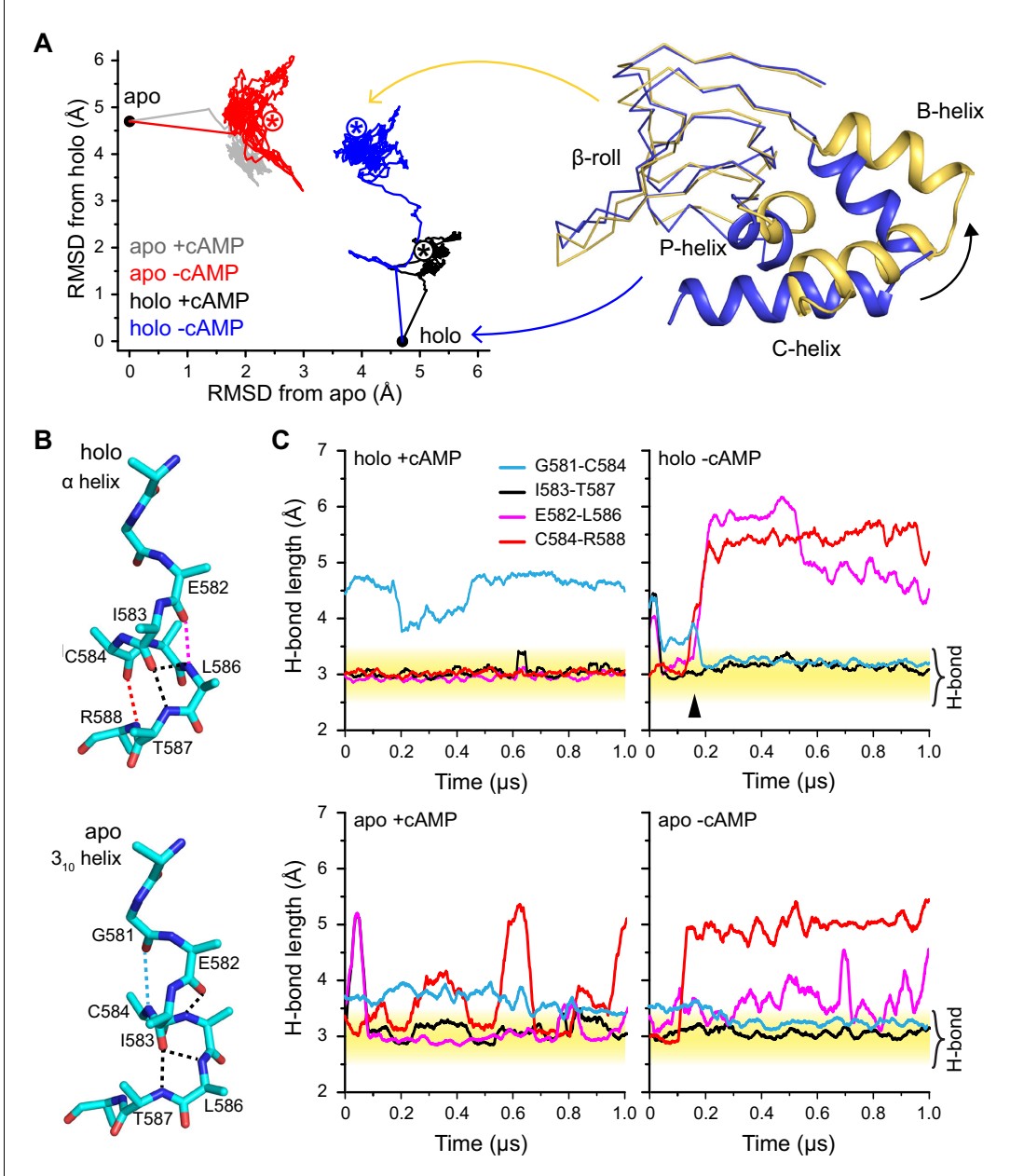

**Figure 5.** Global and local structural features stabilized by cAMP. (**A**) RMSDs from apo (this work) and holo (PDB 3U10) crystal structures for the HCN2CNBD (residues 515–632) during 1 μs simulations starting in either the apo or holo structure both with and without cAMP in the binding site. Structures corresponding to the first (blue) and last (yellow) frames of the simulation for the holo structure with cAMP removed are shown (N-terminal helices omitted for clarity) aligned to the β-roll domain. Removal of cAMP from the holo structure resulted in the B, C and P helices swinging outwards towards their apo positions. (**B**) The P-helix adopts a mostly α-helix or mostly $3_{10}$-helix in the holo and apo crystal structures, respectively. Hydrogen bonds are indicated by dashed lines. (**C**) Carbonyl to amide hydrogen bond distances for select P-helix residues over the course of MD simulations starting in the holo (upper) and apo (lower) form either with (left) or without (right) cAMP. The stable α-helical form in the presence of cAMP transitions to a stable mostly $3_{10}$-helix upon removal of cAMP from the holo structure. Upon addition of cAMP to the apo form, the i/i + 4 interactions associated with the α-helical form become more stable while the i/i + 3 interaction associated with the $3_{10}$ form is weakened.

As shown in *Figure 5*, changing the occupancy of the binding site induces structural transitions at both local and global scales; the fact that these changes are observed at the microsecond scale in unbiased MD simulations indicate that they are tightly coupled to cAMP binding. At the local scale, the most visible trend concerns the structural stability of the P-helix, which adopts a $3_{10}$ helical

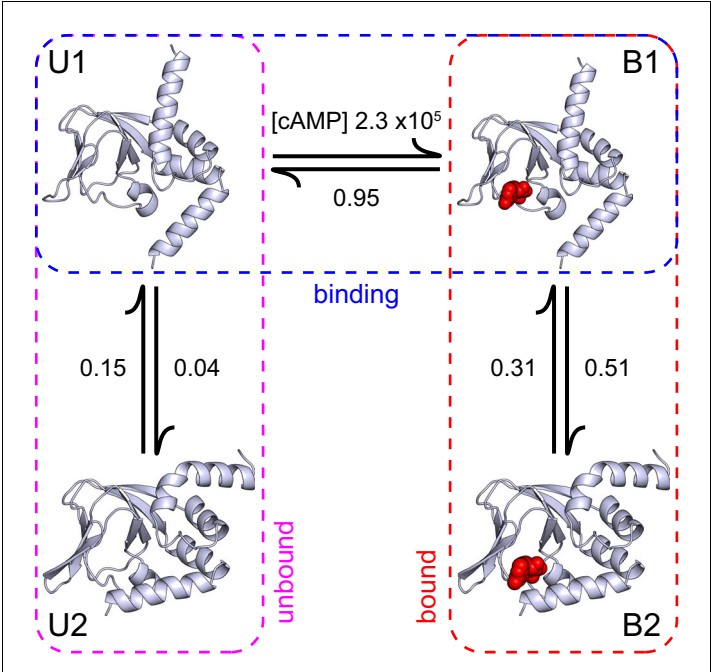

**Figure 6.** A structural model of HCN2 CNBD binding dynamics. A structural model of cAMP (red spheres) binding dynamics at monomeric CNBDs from HCN2 channels. Rate constants ($s^{-1}$ or $M^{-1}s^{-1}$) were optimized using HMM modeling of idealized single-molecule fcAMP binding time series as discussed in the text. Model depicts selective binding of cAMP (horizontal transition) to the apo form of the CNBD (U1) and a subsequent isomerization (vertical transitions) of the ligand-bound CNBD (B1) to its holo form (B2). Isomerization of the unliganded CNBD (U2) prevents cAMP binding due to occlusion of the binding site by the C-helix.

configuration in the apo crystal structure but an $\alpha$-helix in the holo crystal structure. In the holo-cAMP simulation, two key i/i + 4 helical contacts (E582-L586 and C584-R588) are seen to readily break, while an i/i + 3 contact (G581-C584) characteristic of a $3_{10}$ helix is formed; in the holo+cAMP control simulations, the aforementioned i/i + 4 contacts remain stable during the microsecond simulation while the G581-C584 contact is not formed. Similarly, in the apo+cAMP simulation, the two i/i + 4 helical contacts are clearly more stable than in the apo-cAMP control simulation (although the C584-R588 contact constantly breaks and forms with a duration for the bound state spanning from ~10–100 ns), while the G581-C584 contact becomes less stable once cAMP is docked into the apo structure. These consistent trends from independent simulations clearly highlight the direct impact of cAMP binding on the secondary structure of the P-helix.

At the global scale, changing the ligation state in the holo and apo structures induces limited structural transitions during the microsecond simulations, as expected considering the diffusive and slow nature of the U1-B1 and B1-B2 transitions. Nevertheless, it is encouraging that removing cAMP from the holo structure is seen to induce a substantial displacement of the C-terminal $\alpha$-helices (B, C helices) and the P-helix, which all swing outwards away from the binding site towards their positions in the apo structure, similar to the cAMP-dependence of relative distances between many residue pairs observed with FRET and electron paramagnetic resonance (EPR) (*Taraska et al., 2009*; *Puljung and Zagotta, 2013*). Although the holo-cAMP simulation does not correspond to the putative encounter complex in the kinetic scheme, the observed structural relaxation helps reveal the motifs whose stabilization relies most critically on cAMP binding. Such information provides further understanding on the impact of cAMP binding, especially regarding the coupling between local and distal structural transitions.

## Discussion

Typical optical approaches utilizing fluorescent ligands are incapable of resolving low affinity binding events at the single-molecule level due to the large background fluorescence from the high ligand concentrations necessary to drive association. Here, we show that ZMW's in conjunction with smFRET can resolve individual specific binding events in the presence of concentrations of at least 10 µM fluorescent species, extending single-molecule approaches to more physiologically relevant lower affinity binding events than previously possible while eliminating the influence of non-specific binding (e.g. *Figure 2—figure supplement 1*) (*Zhu and Craighead, 2012*). Our observations resolve the dynamics of the distinct events that underlie cyclic nucleotide association in HCN channels, and reveal that cAMP selectively binds to the 'receptive' state of the CNBD, and thereafter induces an isomerization between two distinct bound conformations. As expected given its modular nature (e.g. isolated CNBDs retain similar binding affinities to full length channels and form cAMP-induced tetramers in solution [*Lolicato et al., 2011*]), we show that this isomerization occurs in both monomeric and tetrameric CNBD complexes, and thus is likely to reflect inherent properties of the CNBD in channels. We also successfully crystallized the HCN2 CNBD in a unique unliganded state, which provides a high resolution counterpart to previous ligand-bound structures to aid in structural interpretation of the binding mechanism.

The single-molecule binding data combined with the structural information suggest a model where the initial binding step reflects selective cAMP association to the apo form of the CNBD, and the subsequent isomerization involves a coordinated rotation of the N- and C-terminal α-helices about the rigid β-roll whereby the C-helix caps the bound ligand as in the holo structure (*Figure 6*). HMM modeling suggests that binding occurs selectively from one of two unbound states, which implies that the interchange between these states involves an occlusion of the binding site in the apo form, likely by the C-helix. The slow dynamics for this process imply a relatively large energy barrier that is reduced upon binding cAMP, consistent with EPR observations that the C-helix adopts a cAMP-dependent equilibrium between apo and holo positions (*Puljung et al., 2014*; *Deberg et al., 2016*). Based on both the model predictions that transitions between U2 and B2 states occur either infrequently or not at all, and the structural observation that the C-helix occludes access to the binding site in the holo form, we rule out cyclic models that allow binding to the isomerized state (e.g. model 6 in *Figure 3E*).

Our observations provide the first direct evidence that the binding rate is relatively slow, which suggests that fcAMP binding to the 'receptive' state of the CNBD is rate-limited by additional processes other than simple diffusion, such as ligand reorientation, desolvation and structural rearrangements in the binding site including the P-helix $3_{10}$ to α-helix transition. Thus, the initial encounter complex is either relatively unstable, such that brief encounters shorter than our frame rate were not observed, or it sees a significant barrier to its formation. Regardless, following formation of the initial bound complex the CNBD undergoes a reversible isomerization between two bound forms. Comparison of apo and holo X-ray structures shows that the N- and C-terminal helices are rotated in the isomerized state where the C-helix caps the binding site (*Figure 4*). Thus, the observed isomerization may reflect dynamic movement of the C-helix between apo and holo positions (*Figure 6*). However, we cannot rule out the possibility that the C-helix also moves during the initial binding step; indeed, by perturbing the ligation state of the CNBD in holo and apo structures in otherwise unbiased molecular dynamics simulations at the microsecond scale, we observe that the cyclic nucleotide has a direct impact on the structure of the P-helix and orientation of the B, C helices in the C-terminal region (*Figure 5*). In this case, the subsequent isomerization may involve rearrangements in the N-terminal C-linker, a region that was truncated in order to obtain the apo structure, and therefore for which we have incomplete structural information regarding its unliganded form. Nonetheless, these data establish the dynamics of the distinct events that underlie both selective binding and cAMP-induced conformational changes that regulate the channel pore. We note, however, that although it is highly plausible that fcAMP and cAMP bind with similar dynamics given their similar affinities and functional effects, we cannot rule out the possibility that cAMP association dynamics differ from those observed here for fcAMP.

The large-scale conformational change of the HCN2 CNBD induced by cAMP binding can be viewed as a rotation of the N- and C-terminal helices, taken as a single domain, in relation to the rigid β-roll. Crucially, the C-helix that caps the bound ligand moves in conjunction with the

N-terminal helices connected to the tetramerization module (C-linker) and channel pore. Thus, a reciprocal relationship between ligand binding and tetramerization inevitably follows: ligand binding is expected to shift the equilibrium toward tetramerization and, in turn, anything that promotes tetramerization would be expected to enhance ligand binding. This idea is schematically illustrated in *Figure 4h*, and provides a facile explanation for cAMP-induced tetramerization of monomeric CNBDs in bulk solution and lower affinity for cAMP of CNBDs lacking a C-linker (*Lolicato et al., 2011*).

The notion that ligand activation involves isomerization of a ligand-bound receptor to a 'flipped' or 'primed' configuration has been predicted primarily on the basis of analysis of downstream current recordings (*Lape et al., 2008*; *Mukhtasimova et al., 2009*; *Colquhoun and Lape, 2012*; *Purohit et al., 2014*; *Thon et al., 2015*; *Goldschen-ohm et al., 2014*; *Gielen et al., 2012*). However, these various bound states have previously never been directly observed. Our studies here provide direct evidence for such a binding process, and also shed new light on a long-standing debate about the transition pathways that define the binding mechanism–conformational selection versus induced fit. We find that the initial step involves selective binding to the 'receptive' state (U1), which thereafter undergoes a conformational change that traps the ligand in the bound conformation (B2). Our observations reveal the dynamics and structural detail of the distinct steps that underlie cyclic nucleotide regulation of HCN pacemaker channels, and lay the necessary foundation to probe the molecular details involved in each separate step of multi-subunit complexes. The combination of dynamic and structural observations provide a general approach for revealing the molecular details governing signaling based on weak-binding, a critical and previously inaccessible class of molecular recognition processes.

## Materials and methods

### Cloning and constructs

All cloning and mutagenesis was performed using QuikChange-like protocols as described in detail previously (*Klenchin et al., 2011*). All constructs were sequence-verified over the entire ORF length on both strands. Biotin ligase expression construct was made by amplifying full length BirA gene from *E. coli* genomic DNA and cloning it into a pET21 backbone as an MBP fusion protein. The plasmid backbone contains an unidentified defect that results in about 5-fold lower plasmid copy number than the typical pET vectors. The protein linker sequence between MalE and BirA was SSSSGTA SGGATTSENLYFQGG.

HCN2 fragment was originally obtained as a synthetic DNA (Integrated DNA Technologies) with the sequence that was codon-optimized for expression in *E. coli*. All residue numbering refers to a mouse HCN2 protein. CNBD fragments were always expressed as fusions with an N-terminally 8His-tagged MBP, cloned into a derivative of pET28 plasmid. For protein expression intended for single molecule imaging experiments, the final construct included, after TEV protease recognition site, an AviTag sequence (*Beckett et al., 1999*), flexible linker and thrombin recognition site. The resulting protein thus contained, after cleavage and purification, the sequence GGLNDIFEAQKIEWHEGASGG SSGGSSGGLVPRGS at the N-terminal end of the HCN2 residues D443-N640. HCN2 construct intended for crystallization (residues E494-N640) was fused to MBP through a double alanine linker. The GCN4pLI-HCN2 tetrameric construct was the same as the monomeric one except that it had the following sequence inserted between the end of AviTag (GAS) and the start of flexible linker (GGS) sequences: RMKQIEDKLEEILSKLYHIENELARIKKLLGER

### Protein expression and purification

For co-expression of biotin ligase and HCN2 constructs (monomeric or tetrameric versions), BL21 *CodonPlus*(DE3)-*RIL* cells were sequentially transformed first with the HCN2 construct, selecting transformants overnight on kanamycin/chloramphenicol plates, then with BirA construct, selecting overnight on ampicillin/kanamycin/chloramphenicol plates. Several clones were picked to inoculate 125 ml of MDG medium (*Studier, 2005*) and cultured overnight at 37°C. 30 ml of the resulting culture was used to inoculate 1 L of LB medium in 2 L shake flasks that were grown at 37°C until $OD_{600}$ of about 0.5 (all $OD_{600}$ values refer to measurements done in Beckman DU-640 spectrophotometer), at which point 1 ml of 100 mM solution of biotin in DMSO was added to each flask. After additional

30 min of shaking, the cultures were cooled on ice and induced with 1 mM IPTG. After 20 hr of growth at 16°C, cells from 4 L of culture were pelleted, washed in 1 L of ice-cold 20 mM Tris, 100 mM NaCl and 2 mM EDTA, pH 8.0, the cell paste frozen in liquid nitrogen and stored at −80°C until needed. Expression of the construct for crystallization followed the same outline except that seed culture used was grown at 30°C overnight in MDG with kanamycin/chloramphenicol and no biotin was added before induction at $OD_{600}$ 1.0.

The biotinylated HCN2 constructs containing the entire C-linker sequences were purified as follows. Unless otherwise stated, all procedures were performed at 4°C. 10 g of frozen cells were resuspended in 60 ml of buffer A (20 mM HEPES, 200 mM NaCl, 25 mM imidazole, 0.5 mM TCEP, 10% v/v glycerol, pH 7.5) with an addition of extra 0.5 mM TCEP and protease inhibitors (house-made cocktail equivalent to Roche's 'cOmplete EDTA-free' tablets). The cells were disrupted with ten cycles of sonication on ice-water bath at ~93 W power output while monitoring suspension temperature, keeping cycles short enough to prevent temperature raising above 8°C and resuming at 2–3°C. The suspension was spun for 30 min at 48,000 g and the supernatant was loaded by gravity onto a 6 ml Ni-NTA (Qiagen) equilibrated with buffer A. The column was then washed by gravity with 200 ml of buffer A followed by two 50 ml wash steps with modified buffer A containing higher final imidazole concentrations, 32 and 40 mM. Remaining bound protein was eluted with 18 ml of modified buffer A containing 250 mM imidazole. Approximately 1 mg of TEV protease per 50 mg of eluted protein was added and the mixture was dialyzed overnight against 2 L of 20 mM HEPES, 100 mM NaCl, 0.5 mM TCEP, pH 7.5 in dialysis bags with 15 kDa cut-off (Spectrum Laboratories cat. # 132122). Following dialysis, a precipitate that formed in the case of mutant HCN2s was spun down for 20 min at 48,000 g and the supernatant loaded by gravity onto a fresh 10 ml Ni-NTA column equilibrated with 20 mM HEPES, 200 mM NaCl, 0.5 mM TCEP, 10% glycerol, pH 7.5. After 100 ml wash with the same buffer that removes minor contaminants, the untagged HCN2 was eluted isocratically with 10 mM imidazole in the equilibration buffer. The eluted protein was concentrated by ultrafiltration (Amicon Ultra-15, 10 kDa cut-off) at room temperature to approximately 7 mg/ml and frozen in liquid nitrogen as 0.3 ml aliquots in screw cap tubes.

Purification of the MBP-HCN2 fusion for crystallization followed the same protocol as for biotinylated HCN2 with the following exceptions: purification used 15 g of cells and 100 ml of buffer A; ethylene glycol was used in place of glycerol in buffer A; first Ni-NTA column was 10 ml in volume and the second column was 13 ml; the buffer used during the second Ni-NTA contained 100 mM NaCl and no glycerol. The final protein fraction was concentrated by ultrafiltration (Amicon Ultra-15, 50 kDa cut-off) to approximately 140 mg/ml, frozen in liquid nitrogen as droplets of about 30 μl and stored at –80°C until needed.

Purification of the GCN4pLI-HCN2 tetramer followed identical steps to the monomer up to the elution from the first Ni-NTA column. Thereafter, the eluate (20 ml) was dialyzed against 20 mM HEPES, 100 mM NaCl, 1 mM TCEP, 0.1 mM EDTA, pH 7.5 overnight and dialyzate mixed with an equal volume of 40 mM HEPES, 600 mM NaCl, 20% glycerol, 1 mM TCEP, pH 7.5 before adding ~1:20 of TEV protease by protein mass for cleavage overnight at 4°C. Precipitated HCN2 tetramer was separated from MBP by centrifugation at 2500 g for 15 min and the pellet resuspended in 40 mM HEPES, 600 mM NaCl, 20% glycerol, 2 mM TCEP, 0.1% LDAO (Sigma, cat. no. 40236), pH 7.5 (Buffer B), followed by homogenization in glass-teflon Potter-Elvehjem homogenizer. Following 30 min centrifugation at 40,000 g, the solubilized fraction was loaded onto a fresh Ni-NTA column and the GCN4pLI-GCN2 fusion eluted in the Buffer B containing 20 mM imidazole. The eluate (~13 mg/ml protein) was flash-frozen in liquid nitrogen as 0.6 ml aliquots and stored at –80°C until needed.

In all cases, protein concentration was determined from $A_{280}$ using theoretical extinction coefficient values derived by the Protparam tool (*Gasteiger et al., 2005*).

## Protein labeling

The C508A/C584S/E571C mutant of CNBD was used for all the single molecule fluorescence experiments. To remove TCEP that may interfere with labeling, a 0.3 ml of 7 mg/ml purified biotinylated monomeric protein was buffer-exchanged at room temperature on a 5 ml spin gel-filtration column packed with Bio-Gel P6 (Fine grade, Bio-Rad) equilibrated with degassed buffer C (20 mM HEPES, 200 mM NaCl, 0.2 mM EDTA, 10% glycerol, pH 7.5). To increase the recovery, 0.3 ml of buffer B was additionally applied immediately after protein application to the column. Working as quickly as

possible, the protein was further diluted to 2 mg/ml and mixed with an equal volume of 0.14 mM of maleimide DyLight 650 solution that was freshly prepared by diluting 4.7 mM stock (made with dry DMSO) into degassed buffer B. After two hours of incubation in the dark at room temperature, the reaction was quenched by adding DTT to 20 mM final, the protein was concentrated by ultrafiltration to ~0.5 ml (Amicon Ultra-4, 10 kDa cut-off) and purified from unreacted dye by gel filtration on Superdex 10/300 column using buffer C containing 1 mM DTT. Eluted protein fractions were pooled, mixed with 1/19 th volume of 200 mg/ml BSA solution made in buffer C containing 10 mM DTT and frozen in liquid nitrogen as droplets of about 20 µl that were stored at −80°C until needed. Control experiment with the unreactive C508A/C584S mutant demonstrated that less than 1% of total protein in the sample incorporates the dye following the labeling procedure. This confirms the impression from the crystal structures that the third cysteine, C601, is buried at all times in the beta-roll hydrophobic environment and is not accessible to the hydrophilic reactive dye. The small degree of incorporation that can be seen can be equally due to a small fraction of surface-denatured protein in the mixture as well as some minor cysteine-containing contaminants that are not even seen on a typical protein gel.

The GCN4pLI-HCN2 tetramer was labeled using the identical procedure except that during all dilution, buffer-exchange and purification by gel-filtration steps a degassed and argon-saturated buffer B was used.

## Solution FRET and anisotropy

Bulk solution FRET and anisotropy measurements were performed with a spectrofluorometer (HORIBA Scientific Fluoromax-4) in a buffer consisting of 20 mM HEPES, 100 mM NaCl, 20 mM imidazole, 10% glycerol and 0.5 mM TCEP. Emission spectra of acceptor-labeled CNBD and fcAMP were obtained with 532 nm excitation, 5 nm slit widths for both excitation and emission, 2 nm steps and 100 ms integration time per step. Anisotropy for fcAMP during titration of non-fluorescent CNBD was measured in a 150 µl cuvette for excitation at 530 nm and emission at 565 nm with 2 nm slit widths and an integration time of 10 s. The fraction of bound CNBDs was computed from the anisotropy curve by first subtracting the anisotropy of freely diffusing fcAMP in the absence of CNBD and then fitting the resulting normalized curve to the equation $1/(1+K_d/[CNBD])$, where $K_d$ is the dissociation constant for fcAMP binding.

## Imaging fcAMP binding at single CNBDs in ZMWs

Arrays of ZMWs (Pacific Biosciences) with a biotin-doped PEG layer on their bottom surfaces were first incubated for 2 min in 0.05 mg/ml streptavidin (Prospec, cat # PRO-791), then washed by exchanging the solution volume five times with wash buffer, which consisted of phosphate buffered saline (PBS; pH 7) supplemented with 50 mM NaCl, 2 mg/ml bovine serum albumin (BSA), 1 mM Trolox, 2.5 mM protocatechuic acid (PCA), and either 1 mM TCEP or 2 mM DDT. Next, biotinylated acceptor-labeled monomeric CNBDs were diluted in wash buffer to a working concentration of ~0.1 nM and deposited on ZMW arrays by incubating for 10 min followed by 5–10 solution volume exchanges with wash buffer. This resulted in most ZMWs containing either zero or one CNBD molecule. Finally, the solution volume was exchanged for imaging buffer, which consisted of wash buffer plus 250 nM protocatechuate 3,4-dioxygenase from Pseudomonas sp. (PCD; the oxygen scavenging counterpart to PCA) and various concentrations of 8-(2-[DY-547]-aminoethylthio) adenosine-3',5'-cyclic monophosphate (fcAMP; BioLog). Evaporation of the small buffer volume (~75 µL) bathing the ZMWs within a recording period of up to a couple of hours was essentially abolished by placing a glass coverslip over the top of the array.

ZMW arrays were placed on an inverted microscope (Olympus IX-71) and imaged under either 532 or 640 nm laser excitation (Coherent). Laser power at the sample was 60 W/cm$^2$ at 532 nm and 25 W/cm$^2$ at 640 nm. Lasers were fed into a single AOTF (Laser Launch) that enabled computer control over the excitation time course at each wavelength. The effective observation volume, which is a product of the excitation intensity and emission observation probability, decays exponentially along an axis perpendicular to the sample surface at the bottom of each ZMW. The decay profile depends on both wavelength and ZMW diameter (*Levene et al., 2003*; *Zhu and Craighead, 2012*). Fluorescence emission from 150–200 nm diameter ZMWs first passed through a multiband dichroic and filter cube for imaging Cy3/Cy5 and equivalent dyes (Semrock Brightline Cy3/Cy5-A-OMF), after which

the donor and emission spectra were split with a 650 nm longpass dichroic (Semrock Brightline) and bandpass filtered using pairs of edge filters (donor: 532–632.8 nm; acceptor: 632.8–945 nm; Semrock EdgeBasic and Brighline) and then imaged on two separate 512 × 512 EMCCD cameras (Andor iXon Ultra X-9899) at a frame rate of 10 Hz. Both imaging and alternating wavelength excitation on interleaved frames was controlled with Metamorph software (Molecular Devices).

## Single-molecule ligand binding image analysis

Unless specified otherwise, all analysis was accomplished with custom software programs written in either Matlab (The MathWorks, Natick, MA; https://github.com/marcel-goldschen-ohm), Python or ImageJ. Briefly, for each ZMW array, the simultaneously recorded image time series for donor (fcAMP) and acceptor-labeled CNBD emission intensities were each further split into two sets of interleaved frames corresponding to direct excitation of either the donor or acceptor. The resulting four image time series are denoted $I_{DD}$ (donor emission upon direct excitation), $I_{DA}$ (donor emission upon acceptor excitation), $I_{AA}$ (acceptor emission direct excitation) and $I_{AD}$ (acceptor emission upon donor excitation). The locations of ZMWs in $I_{AA}$ (which are the same in $I_{AD}$) that contained one or more CNBD molecules was obtained from a thresholded mask of the average intensity in the first ~20 frames of $I_{AA}$, which corresponds to a time period where most acceptors remained unbleached. ZMW locations were further refined by fitting a two dimensional Gaussian to the local intensity height map. The corresponding ZMW locations in $I_{DD}$ were obtained by applying a two dimensional affine transformation that mapped the acceptor emission image from one camera to the donor emission image in the other camera. The time-dependent fluorescence at each ZMW was obtained by projecting the average image intensity in a 5-pixel diameter circle centered on the ZMW location throughout each image time series. $I_{DA}$ was indistinguishable from a constant background, and thereby ignored in further analyses. Only those ZMWs that exhibited a single acceptor bleach step in $I_{AA}$ corresponding to one CNBD molecule were considered for further analysis.

The smFRET response $I_{AD}$ for each ZMW was corrected for cross talk due to donor emission in the acceptor channel by subtracting 10% of $I_{DD}$, as determined from donor emission in the acceptor camera in the absence of donor. Next, $I_{AD}$ was baseline corrected by subtracting the mean intensity after acceptor bleaching, and in some cases where it was visually apparent also subtracting a very slow exponential decay (t ~100 s). Binding dynamics were determined by idealization of $I_{AD}$ prior to acceptor bleaching using the software vbFRET (*Bronson et al., 2009*) and allowing for a maximum of two idealized levels. vbFRET accepts as input both a donor ($I_D$) and acceptor ($I_A$) time series, and idealizes the smFRET response given by $I_A/(I_A+I_D)$. However, $I_{DD}$ was not anticorrelated with $I_{AD}$ due to the fact that the donor was on the freely diffusing ligand, and therefore not present in the excitation volume throughout the experiment. Thus, to avoid adding noise during the idealization, $I_{AD}$ was idealized directly by providing vbFRET with $I_A=I_{AD}$ and $I_D= max(I_{AD})-I_{AD}$, which results in a smFRET response that is an arbitrarily scaled copy of $I_{AD}$. The idealized records were overlaid on the raw $I_{AD}$ data series after applying the reverse scaling and baseline shifting, and examined visually. ZMWs with either no idealized events or whose idealization did not pass visual inspection were removed from further analysis. The signal to noise ratio (S/N) for $I_{AD}$ was computed as the ratio of the rescaled idealized amplitude to the RMS noise for all unbound time points. The distribution of S/N for all ZMWs was well described by a gamma function, and ZMWs with S/N < 2 were rejected from further analysis. The idealization and HMM analyses described here were able to reproduce known models given simulated data at this S/N cutoff. The idealized smFRET time series were interpreted as a binary unbound versus bound time-dependent signal.

HMM analysis of single-molecule binding events was performed with QuB (*Qin et al., 2000*; *Nicolai and Sachs, 2013*). Models were globally optimized to simultaneously describe idealized binding time series for all molecules across all tested fcAMP concentrations with a dead time of 200 ms. For each molecule, the first and last event was removed from the analysis to avoid interpretation of events truncated by our recording window or bleaching of the acceptor, respectively. The number of molecules (and binding or unbinding events) that went into the analysis at each fcAMP concentration was: 0.1 µM: 1234 (1808), 0.3 µM: 988 (1950), 1 µM: 2028 (7795), 3 µM: 1064 (4952) and 10 µM: 1085 (2025). This was more than enough events for the analysis to reproduce known models with similar rates from simulated data. The observation time window prior to acceptor bleaching, and hence the number of events per molecule varied stochastically. Nonetheless, inclusion of molecules with few events (i.e. that bleached rapidly) did not grossly distort our modeling results as very similar

rate constants were obtained from the subset of molecules exhibiting 10 or more events prior to bleaching.

Analysis of fcAMP binding at tetrameric CNBDs was performed in a similar fashion to that for monomeric CNBDs, except that excitation was constant at 532 nm and $I_{DD}$ as opposed to $I_{AD}$ was idealized as described above. The presence of single tetramers was assumed based on sparse labeling (conditions where ~5% of ZMWs contained a tetramer were first established by bleach-step analysis of acceptor-labeled tetramers). The number of tetramers (and binding or unbinding events) that went into the analysis was 147 (8554), where the larger average number of events per molecule as compared to the monomer reflects the longer imaging periods due to lack of truncation by acceptor bleaching.

Given the 100 ms duration of each image frame, our analysis is limited to the detection of dynamics on a comparatively slower time scale. Furthermore, for example, missed brief unbound events will cause bound durations to appear longer than they actually are, which likely contributes to some distortion of our reported dwell times. Most importantly, if such an artifact were to underlie our observation of multiple bound durations, then we predict that with increasing fcAMP concentration (and decreasing average unbound duration) we would observe increasingly lengthy apparent bound durations. However, bound time distributions were not obviously concentration dependent, nor were the relative amplitudes or time constants of their maximum likelihood biexponential fits (*Figure 3—figure supplement 1*). Thus, any distortions to the dwell times are minor, and our observations reflect aspects of the cAMP association process. Furthermore, our observed kinetics are roughly consistent with prior predictions of cAMP association rates based on macroscopic observations using patch clamp fluorimetry (*Kusch et al., 2012*).

## Crystallographic procedures

Crystals of the MBP-HCN2 were grown by vapor diffusion at room temperature from a 1:1 mixture of protein at 10 mg/ml (diluted from frozen stock with 5 mM HEPES, 200 mM NaCl, 6 mM maltose, 2 mM TCEP, pH 7.5) and precipitant solution that contained 34–36% dimethyl polyethylene glycol 500 (Sigma cat. # 445886), 240 mM $KNO_3$, 20 mM $MgCl_2$ and 100 mM Bis-Tris, pH 6.0. Typically, 0.2–0.4 ml of protein:precipitant solution was prepared and drops of varying sizes (5–15 μl) were hung over 0.5, 0.75 or 1.0 ml well solutions of precipitant. Microcrystals formed readily in the oil phase overnight but over a few weeks clusters of stacked plates nucleated randomly in some of the drops in the aqueous phase. Individual plate fragments that could by dissected away from the stacks had typical dimensions of $100 \times 200 \times 20$ μm. Crystals were cryoprotected by transferring into a precipitant solution containing 6 mM maltose, 5.5% v/v of 1,6-hexanediol for one minute followed by loop mounting and dipping into liquid nitrogen. Diffraction data were collected at the GM/CA beamline 23-ID-B (Advanced Photon Source, Argonne National Laboratory, Argonne, IL) and processed with HKL2000 (*Otwinowski and Minor, 1997*). Free reflections were selected with PHENIX (*Adams et al., 2010*) using 'use-lattice-symmetry' option. The structure was solved by molecular replacement with two copies of MPB (PDB ID 1ANF) using Phaser (*Mccoy et al., 2007*). The initial model was built by ARP/wARP (*Perrakis et al., 2001*). This was followed by iterative cycles of manual model building in Coot (*Emsley and Cowtan, 2004*) and restrained and TLS refinement in Refmac (*Skubák et al., 2004*). The crystal that gave the highest quality data appears to have been pseudo-merohedrally twinned with a twin fraction of approximately 0.15. To account for this, a twin refinement against amplitudes was included in the Refmac refinement protocol. Large-scale domain motions were analyzed using DynDom (*Girdlestone and Hayward, 2016*). The structural alignments over a rigid β-roll cage excluded P-helix and flexible loops and were performed with CCP4 program Superpose (*Krissinel and Henrick, 2004*) using Cα atoms of residues 533–550, 553–565, 571–581 and 589–606. The alignments over alpha-helical termini used residues ranges 510–531 and 608–632. Unless otherwise stated, the holo structure used in the alignments was PDB 3U10.

## Molecular dynamics simulations

All-atom MD simulations were carried out with NAMD 2.10 (*Phillips et al., 2005*) using the CHARMM36 force field (*MacKerell et al., 1998*; *Best et al., 2012*; *Hart et al., 2012*) in the NVT ensemble at 310 K using Nosé-Hoover and Langevin piston pressure coupling protocols and a one fs time-step. NAMD's Particle Mesh Ewald algorithm was used for electrostatics with a grid spacing of

1 Å. All nonbonded interactions were treated with a 12 Å cutoff and a switching function turned on at 10 Å. For both the apo (this study) and holo (PDB 1Q5O) structures, simulations were performed in the presence or absence of cAMP in the binding site. For the monomeric holo structure, only residues 494 to 634 were included for consistency with simulations of the apo structure which lacked most of the C-linker. The initial apo structure in the presence of cAMP was generated by aligning the backbone atoms of the β-roll for the apo crystal structure to the same atoms in the holo structure by minimizing their RMSD, and then inserting the coordinates of cAMP from the holo structure into the apo structure. The built-in CY35 patch for the CHARMM36 force field was used for the topology of cAMP (*Hart et al., 2012*). The N- and C-termini were neutralized using the ACE and CT3 patches (*Best et al., 2012*), respectively, to mitigate charge effects due to the truncation of the protein sequence. Each system was solvated in TIP3P water with 150 mM KCl using the built-in plugin in VMD (*Humphrey et al., 1996*) and energy minimized for 500 fs with all heavy atoms constrained. Next, constraints on all non-hydrogen atoms other than Cα and cAMP (if present) were released, and an additional 7 ps of energy minimization was performed. With Cα and cAMP still constrained, systems were heated to 310 K over 4 ps, and then equilibrated at constant pressure in the NPT ensemble for 1 ns. The average system dimensions over the final 10 ps were determined and used during the production runs, which were performed in the NVT ensemble as described above. All analysis was performed using VMD (*Humphrey et al., 1996*). Trajectories shown in *Figure 5* were smoothed by applying a running average with 30 ns window.

## Accession numbers

The atomic coordinates and structure factors for the apo HCN2 CNBD have been deposited in the Protein Data Bank (PDB) under the accession code 5JON.

## Acknowledgements

The authors wish to thank Dr Craig Bingman for his kind help in collecting crystallographic data and data processing, James Ng for preliminary single-molecule data acquisition and Dr Michael Sanguinetti for the wild type HCN2 DNA plasmid. This research was supported by funding from the National Institutes of Health to BC (GM084140, NS081293) and DSW (T32 GM007507) and National Science Foundation to RHG (CHE-1254936). BC was also supported by Romnes Faculty fellowship (WARF). GM/CA@APS has been funded in whole or in part with Federal funds from the National Cancer Institute (ACB-12002) and the National Institute of General Medical Sciences (AGM-12006). This research used resources of the Advanced Photon Source, a US Department of Energy (DOE) Office of Science User Facility operated for the DOE Office of Science by Argonne National Laboratory under Contract No. DE-AC02-06CH11357.

## Additional information

### Competing interests

QC: Reviewing editor, *eLife*. The other authors declare that no competing interests exist.

### Funding

| Funder | Grant reference number | Author |
| --- | --- | --- |
| National Science Foundation | CHE-1254936 | Randall H Goldsmith |
| National Institutes of Health | T32 GM007507 | David S White |
| Wisconsin Alumni Research Foundation | Romnes Faculty Fellowship | Baron Chanda |
| National Institutes of Health | GM084140 | Baron Chanda |
| National Institutes of Health | NS081293 | Baron Chanda |

The funders had no role in study design, data collection and interpretation, or the decision to submit the work for publication.

## Author contributions

MPG-O, Conception and design, Acquisition of data, Analysis and interpretation of data, Drafting or revising the article, Contributed unpublished essential data or reagents; VAK, Conception and design, Acquisition of data, Analysis and interpretation of data, Drafting or revising the article; DSW, JBC, Acquisition of data, Analysis and interpretation of data; QC, Supervised, Analysis and interpretation of data; RHG, BC, Conception and design, Analysis and interpretation of data, Drafting or revising the article

## Author ORCIDs

Baron Chanda, ⬛ http://orcid.org/0000-0003-4954-7034

## Additional files

### Major datasets

The following dataset was generated:

| Author(s) | Year | Dataset title | Dataset URL | Database, license, and accessibility information |
|---|---|---|---|---|
| Vadim A Klenchin | 2016 | Crystal Structure of the unliganded form of HCN2 CNBD | http://www.rcsb.org/pdb/explore/explore.do?structureId=5JON | Publicly availble at Protein Data Bank (accession no: 5JON) |

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
