## [Decision Letter]

Thank you for submitting your article "Structure and dynamics underlying elementary ligand binding events in pacemaking channels" for consideration by *eLife*. Your article has been favorably evaluated by Richard Aldrich (Senior Editor) and three reviewers, one of whom, Kenton J Swartz (Reviewer #1), is a member of our Board of Reviewing Editors.

The reviewers have discussed the reviews with one another and the Reviewing Editor has drafted this decision to help you prepare a revised submission.

Summary:

In this paper, Goldschem-Ohm and colleagues investigated binding of cyclic nucleotides and resulting structural changes in the C-terminal region of HCN2 channels. They used a combination of single-molecule FRET with kinetic modeling, X-ray crystallography, and molecular dynamics simulations. The dwell-time distributions with fluorescence cAMP (fcAMP) indicate that the C-terminal fragment exists in two cyclic nucleotide-bound states and two unbound states. The kinetic modeling suggests an allosteric mechanism where the C-terminal fragment exists in two conformations, with cyclic nucleotide-binding exclusively to the "receptive" conformation and stabilizing the conformational change. X-ray crystallography of the apo state of the cyclic nucleotide-binding domain (CNBD) suggests that the conformational change induced by cAMP binding involves a concerted movement of the C-helix and N-terminal helices of the CNBD around the β roll. They present molecular dynamics simulations to support the structures obtained with crystallography.

This is a really excellent paper and contains several exciting advances. The use of zero mode waveguides in combination with single-molecule FRET to examine binding of fcAMP to the CNBD is technically innovative, and allows for quantitative kinetic modeling of fcAMP binding. While the mechanism uncovered is not new, the single-molecule analysis is a rigorous demonstration of the mechanism. The crystal structure of the apo state of the CNBD complements the previously solved NMR and EPR structures and highlights interesting differences between the methods. The molecular dynamics simulations seem to add less insight to the paper compared to the single-molecule FRET and X-ray crystallography experiments.

Essential revisions:

1) The last section of the study, which unfortunately we believe to be its weakness, presents results from MD simulations. The authors report simulations of the apo structure in both the apo state, i.e. U1, and in a state in which cAMP has been modeled into the binding pocket – which, we deduce, is implicitly assumed to resemble B1. The calculations simulate 1 microsecond of dynamics thereafter, seemingly under the expectation that conformational changes leading to the B2 state will occur spontaneously in the presence of cAMP, but not in its absence. Such changes do not occur, but this (negative) result is hardly surprising. In our opinion, that the state simulated is in fact B1, i.e. the assumption that B1 is structurally identical to U1 except that cAMP is inserted in the binding pocket, is not evident; it seems reasonable that as cAMP enters the pocket (a process that is not explicitly simulated here), a significant degree of adaptation is already realized in state B1, at least in the vicinity of the ligand, before the structure transitions to B2. It might well be that the current simulation of U1 with cAMP modeled-in is capturing some of those early adaptations (i.e. the U1 to B1 transition), and that the larger-scale changes (from B1 to B2) will take place in a much longer time-scale – but the manuscript does not include data to support or refute this notion.

Given this negative result, much of the computational data presented derives from simulations of the bound structure, B2. In analogy with the apo-state simulations, the authors simulate this holo-like structure with cAMP bound, as seen in existing X-ray structures, and after removing cAMP from the binding pocket – and in both cases a 1-microsecond trajectory is calculated. The aim of the latter simulation seems to be to observe the return of the CNBD structure to the unliganded form, U1, and indeed this simulation shows changes that are consistent with that state (e.g. the change in pitch of the P-helix). However, what we find problematic here is that the structural state that is simulated, namely B2, does not interconvert with U1 according the kinetic model derived from the single-molecule data. B2 is in equilibrium with B1, which is also a bound state but has a different structure, and so the purpose and interpretation of this simulation in which cAMP is deleted from the B2 state is unclear. (We can imagine how this kind of approach could provide preliminary insights into the nature of the apo-state structure, if this structure was unknown, but this is not the case here.)

In summary, it is unclear to us what questions the authors aimed to address through the specific simulation design presented in the manuscript, as it is not evident that the processes simulated correspond to any one of the conformational equilibria that are said to explain the system. Therefore, it is unfortunately not apparent what value or insight the computational work adds to the experimental study – which we believe to be excellent otherwise. In order to have a more meaningful and compelling computational component, the authors might want to focus on specific aspects of the recognition process, as captured by the existing X-ray structures (e.g. the changes in the P-helix), and design a set of a simulations that conclusively demonstrate a causal connection between cAMP binding and the structural changes observed. Alternatively, the existing simulation data could be reanalyzed/reinterpreted/presented with a different focus, so as to complement the experimental data (e.g. ligand-protein interactions, consistency of fluorophore-labeled bound/unbound ensembles with unlabeled structures, etc.).

---

## [Author Response]

[…]

*Essential revisions:*

*1) The last section of the study, which unfortunately we believe to be its weakness, presents results from MD simulations. The authors report simulations of the apo structure in both the apo state, i.e. U1, and in a state in which cAMP has been modeled into the binding pocket – which, we deduce, is implicitly assumed to resemble B1. The calculations simulate 1 microsecond of dynamics thereafter, seemingly under the expectation that conformational changes leading to the B2 state will occur spontaneously in the presence of cAMP, but not in its absence. Such changes do not occur, but this (negative) result is hardly surprising. In our opinion, that the state simulated is in fact B1, i.e. the assumption that B1 is structurally identical to U1 except that cAMP is inserted in the binding pocket, is not evident; it seems reasonable that as cAMP enters the pocket (a process that is not explicitly simulated here), a significant degree of adaptation is already realized in state B1, at least in the vicinity of the ligand, before the structure transitions to B2. It might well be that the current simulation of U1 with cAMP modeled-in is capturing some of those early adaptations (i.e. the U1 to B1 transition), and that the larger-scale changes (from B1 to B2) will take place in a much longer time-scale – but the manuscript does not include data to support or refute this notion.*

*Given this negative result, much of the computational data presented derives from simulations of the bound structure, B2. In analogy with the apo-state simulations, the authors simulate this holo-like structure with cAMP bound, as seen in existing X-ray structures, and after removing cAMP from the binding pocket – and in both cases a 1-microsecond trajectory is calculated. The aim of the latter simulation seems to be to observe the return of the CNBD structure to the unliganded form, U1, and indeed this simulation shows changes that are consistent with that state (e.g. the change in pitch of the P-helix). However, what we find problematic here is that the structural state that is simulated, namely B2, does not interconvert with U1 according the kinetic model derived from the single-molecule data. B2 is in equilibrium with B1, which is also a bound state but has a different structure, and so the purpose and interpretation of this simulation in which cAMP is deleted from the B2 state is unclear. (We can imagine how this kind of approach could provide preliminary insights into the nature of the apo-state structure, if this structure was unknown, but this is not the case here.)*

*In summary, it is unclear to us what questions the authors aimed to address through the specific simulation design presented in the manuscript, as it is not evident that the processes simulated correspond to any one of the conformational equilibria that are said to explain the system. Therefore, it is unfortunately not apparent what value or insight the computational work adds to the experimental study – which we believe to be excellent otherwise. In order to have a more meaningful and compelling computational component, the authors might want to focus on specific aspects of the recognition process, as captured by the existing X-ray structures (e.g. the changes in the P-helix), and design a set of a simulations that conclusively demonstrate a causal connection between cAMP binding and the structural changes observed. Alternatively, the existing simulation data could be reanalyzed/reinterpreted/presented with a different focus, so as to complement the experimental data (e.g. ligand-protein interactions, consistency of fluorophore-labeled bound/unbound ensembles with unlabeled structures, etc.).*

We thank the reviewers for their critical evaluation of the MD simulation section, and we agree with the reviewers that the motivation of the MD studies in the original draft was not precisely stated. As the reviewers correctly point out, the simulation that showed most structural changes (holo-cAMP) in fact does not describe a process in our kinetic scheme (i.e., B2 and U2 are not directly connected).

In the revision, we have completely rewritten the MD section (“Molecular Dynamics simulations”) to properly put the MD simulations in the context. Briefly, considering the slow time scale associated with the U1-B1 and B1-B2 transitions, and the fact that there is little explicit experimental characterization for the B1 state, we do not anticipate that molecular dynamics (MD) simulations for these transitions are warranted. However, MD simulations are useful for further understanding how the ligand (cAMP) stabilizes the holo structure at both local and global scales. By monitoring structural responses to a change in the ligation state (e.g., holo-cAMP and apo+cAMP), we are able to identify structural motifs that are most directly stabilized by cAMP binding; this is not straightforward to accomplish by examining the static crystal structures alone, especially when allosteric effects are implicated.

We have also toned down the way that MD simulations are interpreted in the Discussion section, and stated that the most relevant observation is that “the cyclic nucleotide has a direct impact on the structure of the P-helix and orientation of the B, C helices in the C-terminal region (revised Figure 5)”. In our opinion, the MD simulations despite all the caveats capture the initial trajectory of the ligand induced conformational transitions and therefore provide a strong support of our model.